# Enhancement of Dissolving Capacity and Reducing Gastric Mucosa Irritation by Complex Formation of Resibufogenin with β-Cyclodextrin or 2-Hydroxypropyl-β-cyclodextrin

**DOI:** 10.3390/molecules27103213

**Published:** 2022-05-17

**Authors:** Nan Liu, Huan-Ping Chen, Zi-Meng Yang, Ming-Yu Xia, Dong Wang, Ling-He Zang, Dong-Chun Liu

**Affiliations:** 1Graduate School of Traditional Chinese Medicines, Shenyang Pharmaceutical University, Shenyang 110016, China; ykdxflj@163.com (N.L.); 13658639203@126.com (H.-P.C.); dongwang@syphu.edu.cn (D.W.); 2Graduate School of Pharmacy, Shenyang Pharmaceutical University, Shenyang 110016, China; 3Wuya College of Innovation, Shenyang Pharmaceutical University, Shenyang 110016, China; dbycyzm0924@163.com; 4Graduate School of Life Science and Biopharmaceutics, Shenyang Pharmaceutical University, Shenyang 110016, China; xmywd@vip.sina.com (M.-Y.X.); linghezang@sina.com (L.-H.Z.)

**Keywords:** resibufogenin, cyclodextrin, inclusion complex, dissolution rate, gastric mucosa irritation

## Abstract

Resibufogenin (RBG) is a natural medicinal ingredient with promising cardiac protection and antitumor activity. However, poor solubility and severe gastric mucosa irritation restrict its application in the pharmaceutical field. In this study, the inclusion complex of RBG with β-cyclodextrin (β-CD) and 2-hydroxypropyl-β-cyclodextrin (HP-β-CD) was prepared using the co-evaporation method, and the molar ratio of RBG to CD was determined to be approximately 1:2 by continuous variation plot for both CDs. The formation of inclusion complexes between RBG and each CD (RBG/β-CD and RBG/HP-β-CD) was evaluated by phase solubility study, Fourier transform infrared spectroscopy, and thin-layer chromatography. Powder X-ray diffraction and differential scanning calorimetry confirmed drug amorphization and encapsulation in the molecular cage for both CDs. Moreover, the inclusion complexes’ morphologies were observed using scanning electron microscopy. The dissolution rate of the inclusion complexes was markedly improved compared to that of RBG, and the complexes retained their antitumor activity, as shown in the in vitro cytotoxicity assay on a human lung adenocarcinoma cancer (A549) cell line. Moreover, less gastric mucosal irritation was observed for the inclusion complex. Thus, the inclusion complex should be considered a promising strategy for the delivery of poorly water-soluble anticancer agents, such as RBG.

## 1. Introduction

Resibufogenin (RBG, 3-hydroxy-14,15-epoxy-20,22-dienolide glycoside (Figure 1)), the main active ingredient of the traditional Chinese drug ChanSu, is isolated from the skin venom gland of Asiatic toad *Bufo gargarizans* [1]; it shows a variety of biological activities, including cardiac protection [2], as a local anesthetic [3], and as an anti-inflammatory agent [4]. In recent years, RBG has been widely used in the treatment of malignancy, such as ovarian cancer [5], colorectal cancer [6], pancreatic cancer [7], and gastric cancer [8]. Moreover, it has been reported to inhibit the growth of human gastric cancer BGC-823 cells and induce cell apoptosis, the underlying mechanism being related to the mitochondrial pathway [9,10]. However, the clinical application of RBG has remained limited due to its poor water solubility, strong toxicity, and gastric mucosal irritation. Various methods have been reported to date to improve the water solubility of RBG, including the formulation of solid dispersions [11] and using liposomes [12]. However, there is no evidence showing any of these methods effectively decrease the gastric mucosal irritation caused by RBG. Thus, the discovery of an efficient and nontoxic carrier for RBG is vital in order to extend its therapeutic applications further.

Cyclodextrins (CDs) are cyclic oligosaccharides consisting of six, seven, or eight D-glucopyranose units (α, β, or γ-CD) linked by α-(1-4) glycosidic bonds [13]. Several driving forces have been proposed for the inclusion of substrates into CDs, including hydrogen binding, van der Waals force, hydrophobic interaction, and the release of “high energy water” molecules from the cavity [14,15]. CDs have been used successfully to improve the solubility, chemical stability, and bioavailability of a number of poorly soluble compounds [16,17,18] and have been applied as pharmaceutical adjuvants for over 20 years [19]. β-CD, the most commonly used CD, has the characteristics of low solubility, easy crystallization, separation, and purification, low cost, and moderate cavity size [20].

Recently, various hydrophilic cyclodextrin derivatives have been developed for a broad range of medical applications [13]. For example, 2-hydroxypropyl-β-cyclodextrin (HP-β-CD), a hydroxyalkyl derivative of β-CD, has been widely used owing to its inclusion ability and high water solubility [21]. Compared to that of β-CD, the solubility of HP-β-CD is >60% (*w*/*w*) in water [22]. Toxicological studies have shown HP-β-CD to be well-tolerated via both intravenous and oral administration [23,24]. Thus, HP-β-CD can be used as a promising drug delivery system with comprehensive applications and greater clinical potential [25,26].

This study aimed to increase the aqueous solubility and dissolution properties of RBG while decreasing the related gastric mucosal irritation and toxicity. Toward that goal, inclusion complexes of RBG were prepared with β-CD and HP-β-CD using the co-evaporation method. Interactions of RBG/CD inclusion complexes (RBG/β-CD and RBG/HP-β-CD) in solution were investigated by phase solubility analysis, and the formation of inclusion complexes was characterized by Fourier transform infrared spectroscopy (FT-IR), powder X-ray diffraction (PXRD), differential scanning calorimetry (DSC), scanning electron microscopy (SEM), and thin-layer chromatography (TLC). Dissolution properties of the inclusion complexes were evaluated relative to those of free RBG and physical mixtures. In addition, an in vitro cytotoxicity assay was performed to determine the bioactivity of RBG/CD inclusion complexes. Finally, KM mice were used to evaluate gastric mucosal irritation due to the inclusion complexes.

## 2. Materials and Methods

Materials. RBG (>98%) was obtained from toad venom and purified in our laboratory, as reported previously [27]. β-CD and HP-β-CD were purchased from Ashland (Ashland Inc., Covington, KY, USA); 3-(4,5-Dimethylthiazol-2-yl)-2,5-diphenyl tetrazolium bromide (MTT) and thin-layer chromatography silica gel (GF254) were purchased from Sigma-Aldrich (St. Louis, MO, USA) and Qingdao Haiyang (China), respectively. All chemicals and solvents used were of analytical grade.

A549 cell culture. The human lung adenocarcinoma cell line (A549) was obtained from the American Type Culture Collection (Manassas, VA, USA). A549 cells were cultured in Dulbecco’s modified Eagle’s medium (DMEM, Gibco) containing 10% (*v*/*v*) fetal bovine serum (FBS), 60 mg/L penicillin, 50 mg/mL streptomycin, and 1% (*v*/*v*) L-glutamine at 37 °C and 5% CO_2_.

Animals. Male KM mice (4–6 weeks old, weighing 18–22 g) were purchased from the Laboratory Animal Center of Shenyang Pharmaceutical University (Shenyang, Liaoning, China). Mice were housed at 25 °C and 40–60% humidity, with a 12-h light/dark cycle, throughout the experimental period. All animal handling procedures were approved by the Institutional Animal Care and Use Committee of Shenyang Pharmaceutical University (SYXK[Liao]2018-0009), and the experimental procedures were in accordance with the national standard of the Laboratory Animal Requirements of Environment and Housing Facilities (GB 14925-2010) and the Institutional Animal Care and Use Committee of Shenyang Pharmaceutical University.

Continuous variation plot. The stoichiometry of RBG to CD was determined using the continuous variation method [28,29]. The total molar concentrations of guest (RBG) and host (β-CD and HP-β-CD) were kept constant, and a mixture of RBG and β-CD or HP-β-CD was prepared with various molar concentration ratios, such as 1:3, 1:2, 2:3, 1:1, 3:2, 2:1, and 3:1; the mix was stirred at 45 °C for 2 h. CD solutions with the corresponding concentrations were used as the blank. Absorbance values of mixed solutions (A_0_) and RBG-intact solutions (A_1_) at 296 nm were investigated (absorbance values of the solvent and CD were deducted), and the difference (ΔA = A_0_ − A_1_) was determined.

Phase solubility study. Phase solubility studies were performed according to the method reported by Higuchi and Connors [30]. Briefly, an excess of drug was added to 10 mL of an aqueous solution of various concentrations of β-CD and HP-β-CD (ranging from 0 to 15 mM) in 25 mL stoppered conical flasks. The obtained suspensions were shaken at 25 °C for 72 h. After equilibration, the excess drugs were removed by filtration using a filter with 0.45 μm pores, and the concentration of drugs was analyzed by high-performance liquid chromatography (HPLC). The assay was performed in triplicate for each drug-CD system. The phase solubility profile was obtained by plotting the solubility of RBG against the concentration of CDs. The apparent stability constants Kc were calculated from phase solubility diagrams with the following equation:(1)$$\mathrm{Kc}=\frac{slope}{S_{0}\left(1-slope\right)}$$
where *S*_0_ is the solubility of RBG in the absence of CDs.

Preparation of RBG/CD inclusion complexes. RBG/CD inclusion complexes were prepared by the co-evaporation method [31]. Briefly, RBG and CDs (1:2 molar ratio) were accurately weighed, dissolved in 20 mL of 50% aqueous ethanol, and then stirred at 400 rpm for 2 h at 50 °C. The solvent was evaporated under reduced pressure using a rotary evaporator at 45 °C. The complexes were then washed thrice with ethyl acetate to remove free RBG and dried under vacuum to obtain RBG/CD inclusion complexes.

Physical mixtures (PMs) of RBG with β-CD or HP-β-CD (1:2 molar ratio) were prepared by accurate weighing followed by homogeneous blending in an agate mortar for 10 min.

### 2.1. Characterization of RBG/CD Inclusion Complexes

FT-IR. FT-IR spectra of RBG, CDs (β-CD and HP-β-CD), their PMs, and RBG/CD inclusion complexes were recorded in the range of 4000–400 cm^−1^ using an IFS-55 spectrometer (Bruker, Germany) with 32 scans at a resolution of 4 cm^−^^1^. Samples were prepared using KBr disks containing 1 mg of the complex in 100 mg of KBr. FT-IR spectra were analyzed using the spectrophotometer software (OPUS 6.0).

PXRD. The PXRD patterns of RBG, CDs (β-CD and HP-β-CD), PMs, and RBG/CD inclusion complexes were recorded using a Philips X’Pert PRO diffractometer (PANalytical, Eindhoven, The Netherlands) with Ni-filtered Cu Kα radiation at a voltage of 40 kV and a current of 40 mA. The scanning rate was 2 min^−1^ over a diffraction angle (2*θ*) range of 5–60°.

DSC. Thermal analysis of RBG, CDs, PMs, and RBG/CD inclusion complexes was performed using a DSC 60 (Shimadzu, Japan). Samples (3 mg) were accurately weighed, sealed in aluminum pans, and heated over a temperature range of 30–250 °C at a rate of 10 °C/min under a nitrogen flow of 40 mL/min.

SEM. Surface morphologies of pure RBG, pure β-CD, pure HP-β-CD, and RBG/CD inclusion complexes prepared by the co-evaporation method were obtained using an S-3400N scanning electron microscope (Hitachi, Tokyo, Japan) with an accelerating voltage of 5 kV. All samples were electrically conductive, due to a thin coat of gold, for 200 s before being examined.

TLC. Five milligrams of each RBG/CD inclusion complex was dispersed into 1.5 mL of ethyl acetate, sonicated for 15 min, and then centrifuged at 4000 r/min^−1^ for 10 min. The supernatant was used as the ethyl acetate extract. The precipitate was dispersed in 10 mL of absolute ethanol, sonicated for 15 min, and centrifuged at 4000 r/min^−1^ for 10 min. The supernatant was used as an absolute ethanol extract. An appropriate amount of RBG was added to 1.5 mL of ethyl acetate as the reference solution. Five microliters of each of the above solutions were dropped onto silica gel GF254 plates; cyclohexane–chloroform–acetone (4:3:3) was used as the developer, after which the plate was taken out and dried at 25 °C, sprayed with 10% sulfuric acid ethanol solution, and baked at 105 °C until the spots appeared clearly.

Dissolution test. Dissolution tests of RBG, PMs, and RBG/CDs were conducted using an apparatus (Tianjin University Electronics Co, Tianjin, China) based on the paddle method. Briefly, powdered samples (equivalent to 3 mg RBG) were added to 900 mL of phosphate buffer (pH 6.8) at 37 ± 0.5 °C with a paddle speed of 100 rpm. At pre-determined time points, 5 mL samples were withdrawn and replaced with dissolution medium, filtered (pore size 0.45 μm), and then determined by HPLC. All experiments were performed in triplicate.

In vitro cytotoxicity study. Exponentially growing cells were seeded in 96-well plates at a density of 5 × 10^4^ cells/mL. After 24-h incubation, the medium was withdrawn, and cells in each well were incubated with various concentrations (0.001, 0.01, 0.1, 1.0, 10, and 100 μM) of RBG/CD inclusion complexes for 48 h. Next, 150 μL of MTT (5 mg/mL in DMEM) was added, and the plates were further incubated for 3 h. The incubation medium was discarded thereafter, and 150 μL of DMSO was added to each well to dissolve the formazan crystals. Absorbance at 492 nm was measured using a SpectraMax M5 Microtiter Plate Luminometer (Salzburg, Austria). The half-inhibitory concentration (IC_50_) values were calculated using GraphPad Prism software. All experiments were performed in triplicate.

Study of gastric mucosal irritation. Male KM mice were randomly divided into five groups (*n* = 3), namely (I) 0.5% sodium carboxymethyl cellulose (CMC-Na, negative control), (II) acetylsalicylic acid (200 mg/kg, positive control), (III) RBG (4 mg/kg), (IV) RBG/β-CD (4 mg RBG/kg), and (V) RBG/HP-β-CD (4 mg RBG/kg). Six hours after oral administration, mice were sacrificed, their stomach immediately excised and cut along the greater curvature of the stomach, and then washed with saline. A 0.5 cm × 0.5 cm size of the stomach wall was extracted, fixed in formalin, and embedded in paraffin for hematoxylin and eosin (H & E) staining.

### 2.2. Analytical Methods

HPLC analysis. HPLC analysis was performed on a Shimadzu LC 2010A (Tokyo, Japan) consisting of an LC-10Avp pump, an SPD 10Avp UV detector, an SCL-10Avp controller, and Shimadzu CLASS-VP 6.12 chromatographic workstation software for data collection. A reverse-phase HPLC column (Diamonsil ODS, 4.6 mm × 250 mm, 5 μm) equipped with a guard column (C18, 4.6 mm × 10 mm) was employed in this study. The mobile phase consisted of acetonitrile–water (50:50, *v*/*v*) and was adjusted to pH 3.2 with phosphoric acid at a flow rate of 1.0 mL/min. The column temperature was maintained at 40 °C, and the detection wavelength was 296 nm, calculated from the UV spectrum (Appendix A). The injection volume was 20 μL, and the retention time was 6.96 min.

Statistical analysis. All results are presented as means ± standard deviation (SD). Student’s *t*-test or one-way analysis of variance (ANOVA) was used to evaluate significance.

## 3. Results and Discussion

### 3.1. Determination of Inclusion Molar Ratio

The inclusion molar ratio is an important parameter for characterizing the properties of the inclusion complex and is closely related to the equilibrium constant during the inclusion process. In this study, the continuous variation method was used to determine the inclusion ratio [28,29]. The results of the inclusion ratio of RBG and each CD are shown in Table 1 and Table 2, respectively. The maximum ΔA value corresponding to the molar ratio was 1:2 for both the RBG/β-CD and RBG/HP-β-CD systems, indicating that both RBG/β-CD and RBG/HP-β-CD can form inclusion complexes with the same molar ratio of 1:2. It is suggested that when RBG entered the hydrophobic cavity of CD, its physical and chemical properties were changed. Especially when the chromophore entered the cavity of CD, the high electron density in the cavity induced the movement of RBG electrons, causing the change of absorbance. Therefore, the molar ratio corresponding to the maximum absorbance change ΔA value should be the inclusion ratio of the inclusion complex.

### 3.2. Phase Solubility Study

The interaction between the drug and CD is the primary factor in improving the drug’s solubility since it provides not only the solubilizing ability of CD but also the stability constant of the inclusion complex by analyzing the solubility curves [32]. The phase solubility curves of RBG in β-CD and HP-β-CD at 25 °C are shown in Figure 2. The solubility of the RBG in different media is shown in Appendix A. The solubility of RBG increased at lower concentrations of β-CD until it reached the maximum value. In addition, the solubility of RBG was significantly increased by forming an inclusion complex with β-CD and HP-β-CD in water. When the concentration of β-CD was greater than 12 mmol/L, the inclusion complexes achieved limited aqueous solubility and precipitated in aqueous media; thus, the RBG concentration decreased. Conversely, for the HP-β-CD system, the solubility of RBG increased nonlinearly and deviated from the straight line in the negative direction as the HP-β-CD concentration was increased, indicating the formulation of a soluble inclusion complex. According to the Higuchi and Connors classification [33], the diagrams obtained could be classified as B_S_-type and A_N_-type for β-CD and HP-β-CD [34], respectively. Additionally, the stability constants (Kc) estimated from the slope of the phase solubility plots of the RBG/β-CD and RBG/HP-β-CD systems (Appendix A) were found to be 9581 ± 360 M^−1^ and 10,765 ± 492 M^−1^, respectively. The stability constant of the inclusion complex formed between RBG and HP-β-CD was greater than that formed with β-CD, indicating that the inclusion complex formed between RBG and HP-β-CD was more stable than with β-CD, which could be due to the greater water solubility, higher wetting and complexing ability to RBG of HP-β-CD. Stella et al. reported that the stability constant Kc of drugs and CD is generally between 100 and 20,000 mol/L [35]. The Kc value measured in this experiment was relatively large, indicating that the interaction between RBG and the CDs was relatively stable. However, a greater Kc value is not necessarily more desirable since much higher Kc values could negatively affect drug absorbability. Combined with the inclusion stoichiometry of 1:2 for RBG and CDs, the possible inclusion mode for the RBG/β-CD and RBG/HP-β-CD complexes is illustrated in Figure 3.

### 3.3. Characterization of RBG/CD Inclusion Complexes

#### 3.3.1. Fourier Transform Infrared Spectroscopy (FT-IR)

Improvement of the physicochemical properties of RBG, such as hydrogen-bonding interactions with CDs, can effectively increase the water solubility of RBG after the formation of inclusion complexes [36]. Several characteristic peaks were observed in the FTIR spectra of RBG (Figure 4A), including prominent absorption bands of O-H at 3419 cm^−1^, C=O stretching vibration at 1719 cm^−1^, and C-H stretching vibration at 2937 cm^−1^ on the aromatic ring. Additionally, the spectra of CDs showed broader absorption bands at 3500–3300 cm^−1^ due to -OH stretching vibration, another band at 2930 cm^−1^ was attributed to C-H stretching vibration, and the stretching vibration of C-O was observed at 1150–1030 cm^−1^. In the case of PMs, the intensity of all bands decreased, and the superimposition of individual patterns of RBG and CDs was easy to observe, indicating that there were no interactions between the drug and CDs. However, all the RBG absorption bands were covered by the CDs bands, except that the C=O vibration (1719 cm^−1^) of RBG was shifted to lower frequencies of 1707 cm^−1^ and 1708 cm^−1^ in the spectra of RBG/β-CD and RBG/HP-β-CD inclusion complexes (Appendix A), respectively. Such behaviors may be ascribed to the molecular encapsulation of RBG into the cavity of β-CD or HP-β-CD.

#### 3.3.2. Powder X-ray Diffraction (PXRD)

The formation of the RBG/β-CD inclusion complex and the RBG/HP-β-CD inclusion complex was further evaluated by PXRD measurements. As shown in Figure 4B, the characteristic peaks of RBG appeared at diffraction angles (2θ) of 7.50, 15.58, 16.66, 18.83, 19.43, 20.55, 22.35, and 24.23°, indicating a crystalline structure. β-CD exhibited many characteristic crystalline peaks at diffraction angles of 9.09°, 12.73°, 22.71°, and 32.01°, while HP-β-CD was amorphous, displaying a diffuse halo pattern [37,38]. The PMs showed a mixture of peaks divided from crystalline RBG and β-CD, which could be regarded as a simple superposition of the characteristic diffraction peaks of the two compounds. Contrastingly, the RBG/β-CD inclusion complex showed no crystalline peaks of either RBG or β-CD, and new diffraction peaks were observed, indicating the successful formation of the inclusion complex. Moreover, the PXRD patterns of the RBG/HP-β-CD inclusion complex existed in an amorphous state, which did not show any crystalline peaks. These results confirmed that RBG/CD inclusion complex formation led to the amorphous transformation of RBG, indicating the interaction between RBG and CDs [39].

#### 3.3.3. Differential Scanning Calorimetry (DSC)

The thermal properties of the RBG/CD inclusion complexes were investigated using DSC. As shown in Figure 4C, the DSC curve of RBG exhibited an endothermic peak at 168 °C—corresponding to its melting point. The thermal curves of β-CD and HP-β-CD exhibited a broad endothermic peak at 92 °C and 66 °C [40,41], respectively, which could be attributed to the release of crystal water from the CD molecules. Contrastingly, the endothermic peak of RBG along with a broad endotherm of β-CD or HP-β-CD could be observed from PMs with less intensity, which implied the presence of either weak or no interactions between RBG and CDs. Regarding the inclusion complex groups, the endothermic peak of RBG completely disappeared, and the endothermic peak of the inclusion complex was coincident with that of CDs, indicating that RBG was included in the cavity of the CDs [42].

#### 3.3.4. Scanning Electron Microscopy (SEM)

The surface textures of the RBG, CDs, and inclusion complexes are shown in Figure 5. RBG appeared as irregularly shaped crystals, β-CD existed in a polyhedral shape with large dimensions, while HP-β-CD was observed as spherical particles with cavity structures. The RBG/β-CD and RBG/HP-β-CD inclusion complexes appeared to be quite different from β-CD, HP-β-CD, and RBG in terms of size and shape, and the original morphology of RBG and both the raw CD materials disappeared. This phenomenon is considered indicative of the crystalline structure difference between free CDs and the inclusion complexes.

#### 3.3.5. Thin-Layer Chromatography (TLC)

TLC is an adsorption chromatography technique used to isolate compounds in mixtures. Different compounds move at different rates according to their specific solubilities in the mobile phase. The TLC results of the RBG/β-CD and RBG/HP-β-CD inclusion complexes in different solvents—ethanol and ethyl acetate—are shown in Figure 6. The RBG reference and both the inclusion complexes in ethanol possessed fluorescence, which was derived from RBG. Additionally, the R_f_ (rate of flow) value was used as a quantitative indicator of RBG mobility in the mobile phase. The average R_f_ values for both the RBG reference and inclusion complexes in ethanol were the same (0.49), which could be due to the inclusion complex having dissolved in ethanol [43,44]. On the other hand, no spot was noted in ethyl acetate because RBG was encapsulated into the cavity for both CDs and no decomposed compound occurred in ethyl acetate [45]. Combined with the above FTIR, PXRD, DSC, SEM, and TLC results, all these results confirmed that RBG formed inclusion complexes with β-CD and HP-β-CD, and no complex was formed by their physical mixture.

### 3.4. Dissolution Characteristics of the RBG from Each Inclusion Complex

Since RBG is a poorly water-soluble drug, to test whether the inclusion complexes can improve dissolution behavior, dissolution properties in PBS (pH 6.8) were evaluated. The RBG dissolution profiles of crystalline RBG, PMs, and the inclusion complexes of RBG/β-CD and RBG/HP-β-CD are shown in Figure 7. The dissolution rate of the crystalline RBG was very slow, and the cumulative dissolution rates were only 18% and 31% at 60 min and 240 min, respectively. The dissolution rate of the PMs was slightly higher than that of crystalline RBG, and the cumulative dissolution rate reached 50% and 53% at 240 min for RBG/β-CD and RBG/HP-β-CD, respectively. This could be attributed to the improved drug wettability of PMs due to the presence of hydrophilic cyclodextrin, which can reduce the interfacial tension between poorly water-soluble drugs and dissolution medium [46,47]. However, the inclusion complexes exhibited much faster dissolution rates, as evidenced by the fact that approximately 99% of the RBG from inclusion complexes were dissolved within 5 min, while only about 50% and 30% of RBG were dissolved from PMs and crystalline RBG within 4 h, respectively. These results were consistent with the enhanced solubility of RBG shown in the phase solubility studies (Figure 2). Additionally, the dissolution improvement could be largely attributed to the amorphization of RBG after complexation, as indicated by the PXRD and DSC results (Figure 4), and the increased wettability likely contributed to the enhanced dissolution of the inclusion complex. Moreover, the surfactant properties of CDs could also reduce the interfacial tension between the drug and dissolution medium, which increases the dissolution rate of the inclusion complex [48]. Therefore, the RBG/β-CD and RBG/HP-β-CD inclusion complexes can be used as an effective formulation strategy to enhance the solubility and dissolution rate of poorly water-soluble RBG.

### 3.5. In Vitro Cytotoxicity Study

The in vitro cytotoxicity of the RBG/CD inclusion complexes on A549 cells was evaluated using the MTT assay, which relies on the mitochondrial activity of cells and reflects their metabolic activity. Growth inhibition and bioactivity of RBG and the inclusion complexes against A549 cells were investigated using 5-FU as the positive control. The IC_50_ values of RBG and the inclusion complexes are shown in Table 3. Free RBG displayed superior cytotoxicity in A549 cells compared to 5-FU. Moreover, both the RBG/β-CD and RBG/HP-β-CD inclusion complexes exhibited slightly higher cytotoxicity than free RBG, which might be due to the increased solubility of RBG. Our results clearly indicated that the cytotoxicity of both the inclusion complexes was close to that of free RBG and retained the antitumor cytotoxicity of RBG.

### 3.6. Evaluation of Gastric Mucosal Irritation

The safety of the inclusion complexes was evaluated using a gastric mucosa irritation study. As shown in Figure 8A, the structure of gastric mucosal epithelial cells was clear and complete in the negative control group. Contrastingly, many red hyperemia spots, edema, and inflammatory cell infiltration were observed in the positive control group (Figure 8B). After oral administration of RBG, the exfoliation of epithelial cells in the gastric mucosa was observed (Figure 8C). Conversely, almost no small red hyperemia spots were observed after oral administration of the RBG/CD inclusion complexes (Figure 8D,E). One possible explanation for these findings is that RBG in the inclusion complexes exhibited an amorphous crystalline form, which prevented gastric mucosal damage caused by drug crystallization [49]. Additionally, the reduced stomach irritation was probably due to the decreased chance of contact between RBG and the stomach because of the formation of inclusion complexes. Therefore, the CD inclusion complexes can protect the gastric mucosa from gastric injury induced by RBG.

## 4. Conclusions

In this study, RBG/β-CD and RBG/HP-β-CD were successfully prepared by the co-evaporation method at a molar ratio of 1:2, resulting in B_S_-type and A_N_-type phase solubility diagrams, respectively, and confirmed by FT-IR, DSC, PXRD, SEM, and TLC methods. RBG existed in an amorphous state in the inclusion complexes, which significantly improved the solubility compared with the RBG intact. The dissolution rate of RBG was significantly enhanced by encapsulation in the CD cavity. An in vitro cytotoxicity study showed that both inclusion complexes retained the antitumor activity of RBG. Moreover, the inclusion complexes can protect the gastric mucosa against gastric injury induced by RBG. These results may provide guidelines for the development of potential formulations of RBG for use in pharmaceutical fields.

## Figures and Tables

**Figure 1 molecules-27-03213-f001:**
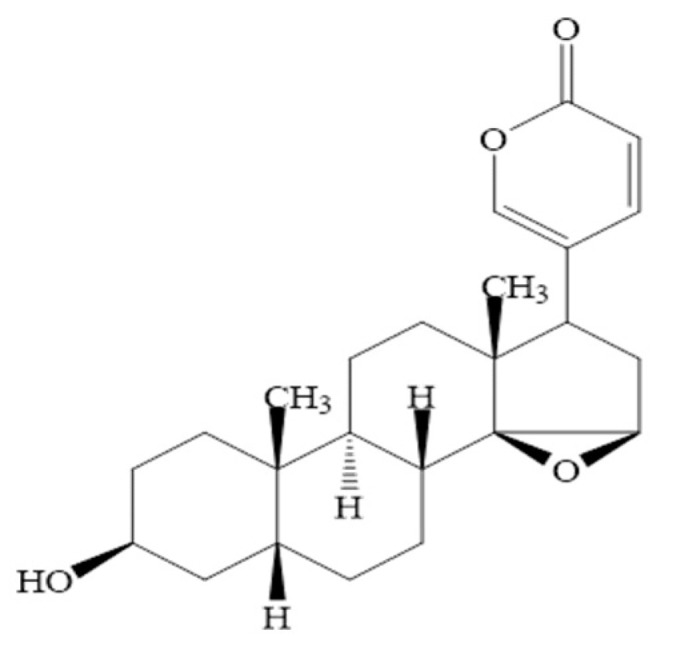
The chemical structures of resibufogenin (RBG).

**Figure 2 molecules-27-03213-f002:**
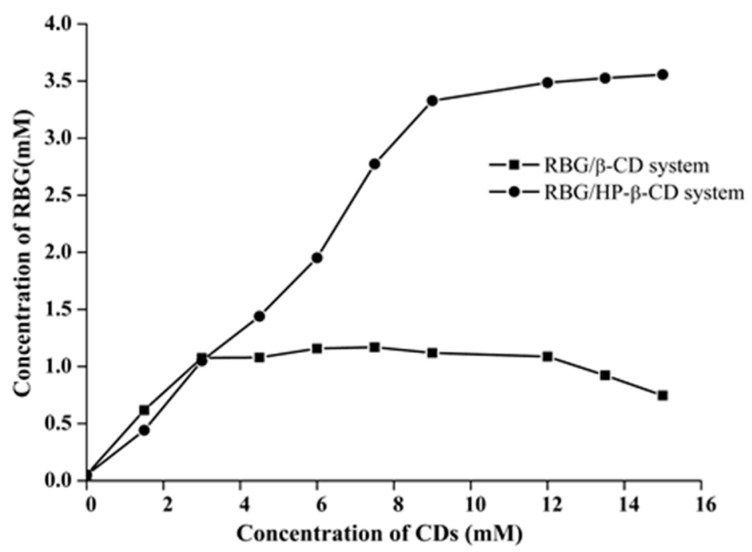
Phase solubility diagrams of RBG/β-CD and RBG/HP-β-CD system. The CD concentration was in the range of 0–15 mM.

**Figure 3 molecules-27-03213-f003:**
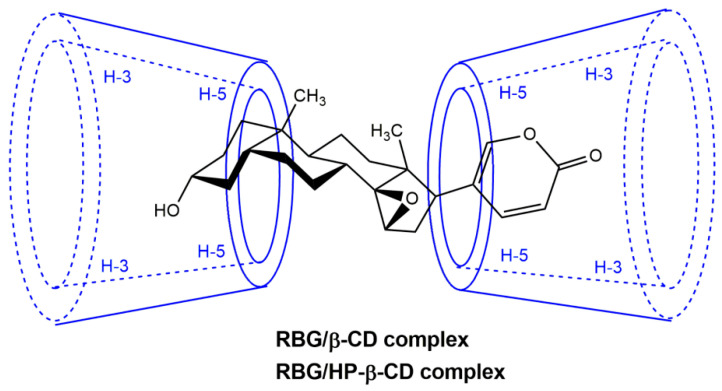
Possible inclusion mode of the inclusion complexes.

**Figure 4 molecules-27-03213-f004:**
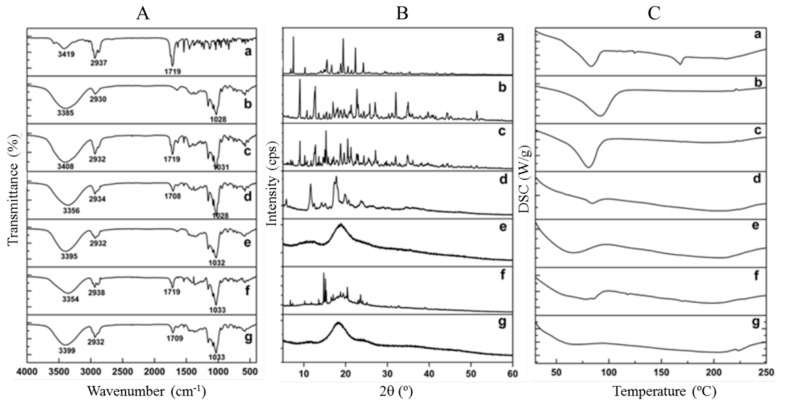
(**A**) FT-IR spectra, (**B**) PXRD patterns and (**C**) DSC curves of (a) RBG, (b) β-CD, (c) RBG/β-CD PM (1:2 molar ratio), (d) RBG/β-CD complex, (e) HP-β-CD, (f) RBG/HP-β-CD PM (1:2 molar ratio), and (g) RBG/HP-β-CD complex.

**Figure 5 molecules-27-03213-f005:**
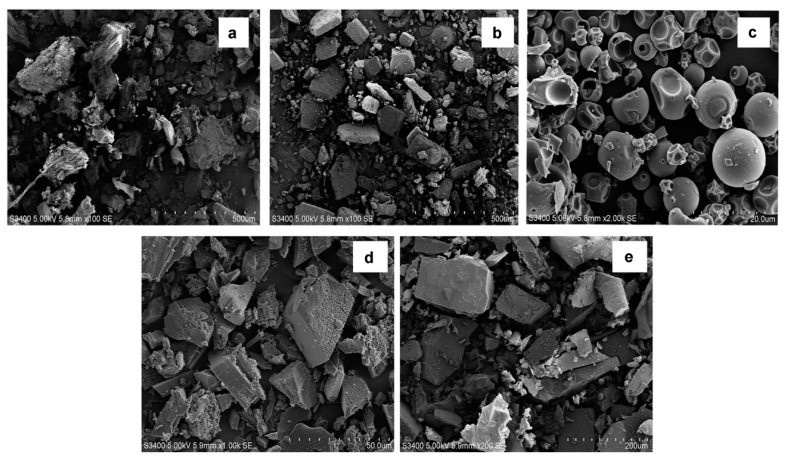
SEM of (**a**) RBG, (**b**) β-CD, (**c**) HP-β-CD, (**d**) RBG/β-CD complex, and (**e**) RBG/HP-β-CD complex.

**Figure 6 molecules-27-03213-f006:**
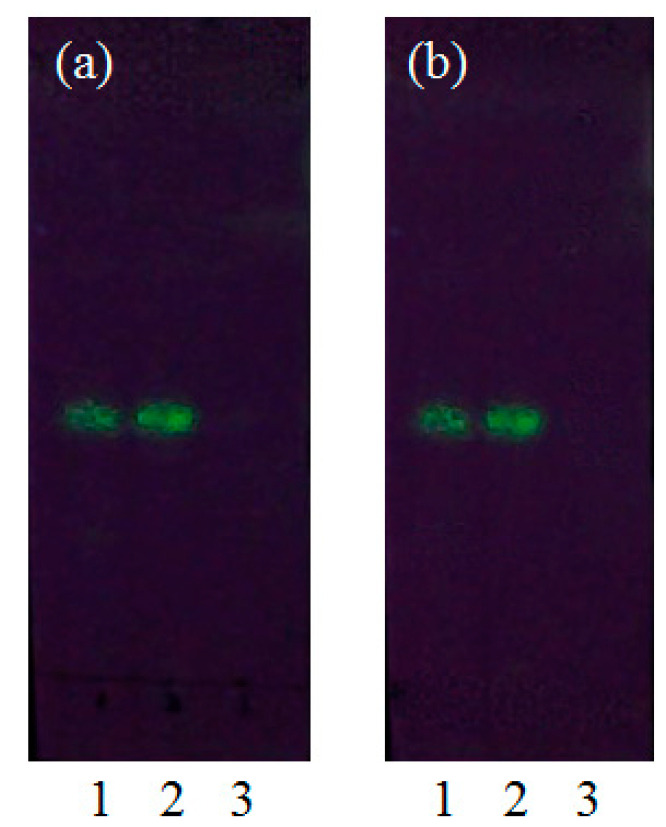
TLC analysis of RBG/β-CD (**a**) and RBG/HP-β-CD (**b**) inclusion complex (1—reference solution; 2—ethanol extract; 3—ethyl acetate extract).

**Figure 7 molecules-27-03213-f007:**
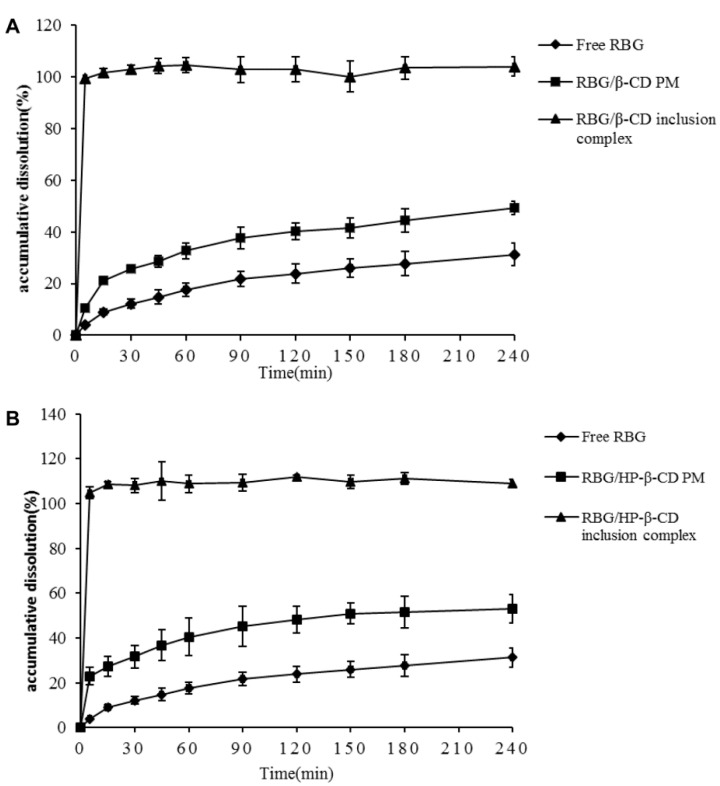
Dissolution profiles of RBG, RBG/β-CD PMs, RBG/β-CD inclusion complexes (**A**), RBG/HP-β-CD PMs, and RBG/HP-β-CD inclusion complexes (**B**) in PBS (pH 6.8) at 37 °C.

**Figure 8 molecules-27-03213-f008:**
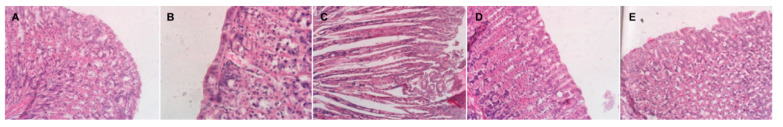
Pathological section of stomach (×10) after oral administration of (**A**) CMC-Na, (**B**) acetylsalicylic acid, (**C**) RBG intact, (**D**) RBG/β-CD, and (**E**) RBG/HP-β-CD.

**Table 1 molecules-27-03213-t001:** Molar ratio experiments of RBG/β-CD system.

No	Molar Ratio	A_0_	A_1_	ΔA
1	1:3	0.108	0.107	0.001
2	1:2	0.131	0.107	0.024
3	2:3	0.116	0.107	0.009
4	1:1	0.104	0.107	−0.003
5	3:2	0.156	0.158	−0.002
6	2:1	0.115	0.108	0.007
7	3:1	0.159	0.158	0.001

**Table 2 molecules-27-03213-t002:** Molar ratio experiments of RBG/HP-β-CD system.

No	Molar Ratio	A_0_	A_1_	ΔA
1	1:3	0.121	0.121	0.000
2	1:2	0.147	0.121	0.026
3	2:3	0.130	0.121	0.009
4	1:1	0.120	0.121	−0.001
5	3:2	0.172	0.168	0.004
6	2:1	0.115	0.121	−0.006
7	3:1	0.166	0.168	−0.002

**Table 3 molecules-27-03213-t003:** In vitro cytotoxicity of RBG and inclusion complex on A549 cells.

	IC_50_ (μΜ)
RBG	0.66
RBG/β-CD	0.58
RBG/HP-β-CD	0.48
5-FU	67.19

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
