# Peer review of "Enhancement of Dissolving Capacity and Reducing Gastric Mucosa Irritation by Complex Formation of Resibufogenin with β-Cyclodextrin or 2-Hydroxypropyl-β-cyclodextrin"

_molecules, 2022, doi:10.3390/molecules27103213_

Round 1

Reviewer 1 Report

In this manuscript the authors show an increase of the solubility of resibufogenin and a decrease of the gastric mucosal irritation by the formation of inclusion complexes with βCD and HPβCD.  According the authors, stability constants are 9763 and 11581 M-1 for βCD and HPβCD, respectively (the standard deviation must be included). Moreover, the results of TLC, dissolution profiles, in vitro cytotoxicity and evaluation of gastric mucosal irritation are similar for βCD and HPβCD, although different behaviour is observed in phase solubility diagrams. There is no discussion about this. After reading this manuscript, I think that a reader could conclude that similar results were obtained with both βCD and HPβCD. From my point of view, this work mainly shows the results obtained with both CDs, however there is no explanation about the effect of the hydroxypropyl groups.

The interest of this manuscript is the effect on the properties of resibufogenin when inclusion complexes are formed. So, it is important to establish conclusively that inclusion complexes have been formed and to explain the effect of hydroxypropyl groups on the results. In my opinion, there is no conclusive evidence that the substrate indeed is entering the cavity of the cyclodextrins. Besides, the possible inclusion mode proposed in figure 3 is only speculative. There are no results in this manuscript to support this inclusion mode.

High values of stability constants were obtained. Did the authors consider the stoichiometry 1:2 to obtain these values?. The equations must be included.

FTIR spectra shown in figure 4 are very small. The change from 1719 to 1709 cm-1 is practically the only one observed for resibufogenin and it is very minor. This change is not appreciated even with a printed figure in full size. If the authors' goal is to demonstrate the formation of the inclusion complex using FTIR, the bands involved must be shown with better resolution.

PXRD results could be a support for RBG/βCD inclusion complex. However, PXRD patterns shown for RBG/HPβCD inclusion complex is practically the same of HPβCD. So, these results are not conclusive. Concerning to DSC results, the authors comment that the dissapareance of the endothermic peak of RBG at 168 °C is indicative of the inclusion complex. This is reasonable, but why a peak associated to the decomposition of the inclusion complex is not oberved?. Besides, the peaks associated to the dehydration of the cyclodextrins are observed at the same temperature for CDs and inclusion complexes. There are no clear explanations about these points.  SEM results are interesting, but the authors must indicate clearly if optical images of free RBG and CDs were obtained with solids that were subjected to the same procedure used for the formation of the inclusion complex, i.e. co-evaporation method. Why this method was chosen?

Additionally:

Absorbance spectra for the determination of stoichiometry must be included in supplementary.

The chromatograms for phase solubility study must be included in supplementary.

Ref 18 is incomplete.

Ref 26 is not adequate. The pioneering work of Higuchi and Connors published in 1964 is: T. Higuchi, K.A. Connors, in: C.N. Reilly (Ed.), Advances in Analytical Chemistry and Instrumentation.

Ref 27 is incomplete. Besides, this reference is not adequate because AN type is not mentioned in this reference.

Lines 110-111: “Phase solubility studies were performed according to the method reported by Higuchi and Connors43,44”. However, ref 43 and 44 are no articles published by Higuchi and Connors.

Reviewer 2 Report

This work is well designed and performed and was easy to read and follow. Below are a few points that the authors might need to take into considerations: 

  • It would be useful to specify the detection limit of the HPLC method, namely that the drug concentration seems to be low at the few first release points.
  • I don't agree that the changes in the powders morphology, shown in the SEM images, prove that the inclusion complex was formed. Similarly, the loss of crystallinity demonstrated by the DSC and XRD analysis do not necessarily mean that the inclusion complex was formed as you may have obtained a simple solid dispersion. NMR analysis might be more conclusion. Please comment on the above mentioned points. 
  • If the drug is administered orally, wouldn't be more bio-relevant to test the drug dissolution in acid medium (gastric) instead of phosphate buffer?

Reviewer 3 Report

The paper by Nan Liu et al. regards the enhancement of dissolving capacity and reducing gastric mucosa irritation by complex formation of resibufogenin with β-cyclodextrin or 2-hydroxypropyl-β-cyclodextrin. The topic is relevant, and the results are interesting and well presented. Hence, in principle, the manuscript deserves publication on Molecules. Nevertheless, the following minor revisions are required:

1) Lines 56-58. Considering the relevance of the topic, other references should be added.

As example, at line 56, the following reference could be added to ref. [14]:

- Crupi, V., Guella, G., Majolino, D., Mancini, I., Rossi, B., Stancanelli, R., Venuti, V., Verrocchio, P. & Viliani, G. T-dependence of the vibrational dynamics of IBP/diME-β-CD in solid state: A FT-IR spectral and quantum chemical study. J. Mol. Struct. 972, 75-80 (2010).

Furthermore, at line 58, the following reference could be added to ref. [15,16]:

- Cannavà, C., Crupi, V., Guardo, M., Majolino, D., Stancanelli, R., Tommasini, S., Ventura, C.A. & Venuti, V. Phase solubility and FTIR-ATR studies of idebenone/sulfobutyl ether β-cyclodextrin inclusion complex. J. Incl. Phenom. Macrocycl. Chem. 75, 255-262 (2013).

2) Line 68. The use of HP-β-CD as a promising drug delivery system is widely reported in literature. Please add some references.

3) Line 83. The structure of β-CD and HP-β-CD is well known. Fig. 1b should be removed.

4) Line 253. “In the case of PMs, the intensity of all bands decreased”. Could the authors please explain why?
